# Label-free characterization of organic nanocarriers reveals persistent single molecule cores for hydrocarbon sequestration

Terry McAfee[1,2], Thomas Ferron[1], Isvar A. Cordova [2], Phillip D. Pickett [3], Charles L. McCormick[3], Cheng Wang[2✉] & Brian A. Collins [1✉]

Self-assembled molecular nanostructures embody an enormous potential for new technologies, therapeutics, and understanding of molecular biofunctions. Their structure and function are dependent on local environments, necessitating in-situ/operando investigations for the biggest leaps in discovery and design. However, the most advanced of such investigations involve laborious labeling methods that can disrupt behavior or are not fast enough to capture stimuli-responsive phenomena. We utilize X-rays resonant with molecular bonds to demonstrate an in-situ nanoprobe that eliminates the need for labels and enables data collection times within seconds. Our analytical spectral model quantifies the structure, molecular composition, and dynamics of a copolymer micelle drug delivery platform using resonant soft X-rays. We additionally apply this technique to a hydrocarbon sequestrating polysoap micelle and discover that the critical organic-capturing domain does not coalesce upon aggregation but retains distinct single-molecule cores. This characteristic promotes its efficiency of hydrocarbon sequestration for applications like oil spill remediation and drug delivery. Such a technique enables operando, chemically sensitive investigations of any aqueous molecular nanostructure, label-free.

[1] Department of Physics and Astronomy, Washington State University, Pullman, WA, USA. [2] Advanced Light Source, Lawrence Berkeley National Laboratory, Berkeley, NC, USA. [3] School of Polymer Science and Engineering, University of Southern Mississippi, Hattiesburg, MS, USA. ✉email: cwang2@lbl.gov; brian.collins@wsu.edu

A grand challenge in medicine and other technologies is resolving the interactions and ordering of biological or synthetically designed supramolecular structures[1–5]. For this reason, polymeric micelles have become increasingly utilized for many applications, including targeted drug delivery[1,2,6–9], environmental remediation[10,11], and engineered molecular lattices[12,13]. Self-assembly originates from linking two polymer species, often one hydrophilic and the other hydrophobic for aqueous applications. In water, the hydrophobic segments aggregate into nanoparticles surrounded by a hydrophilic shell or corona, enabling the capture of other molecules within the core. Some micelles possessing additional stimuli-responsive triggers embedded into their structure behave as molecular machines[1,14,15]. Such technologies are of great current interest, both fundamentally and clinically, for targeted cancer therapies[16–19]. It is clear that the size and shape of the nanoparticles critically impact the properties and performance of such materials[20,21].

One category of these molecules is polysoaps, which contain a random distribution of hydrophobic moieties along a water-soluble backbone[11,22,23]. Their main advantage is the ability to form single-chain (unimeric) micelles without a critical micelle concentration, making possible their use as nanodispersants under highly dilute conditions for environmental/water treatment and targeted drug delivery applications[10,11,24]. However, the internal structure is not well understood for polysoaps, in particular, whether micellar domains coalesce when aggregated or retain separate, high surface area unimeric cores. These distinctions are important in governing sequestration efficiencies. For example, if applied in large concentrations at an oil spill location (typically at the wellhead), micelles will initially be aggregated in solution and will remain so until the micelles are agitated. Whether unimeric cores are retained at both high and low concentration determines the polysoap's effectiveness for hydrocarbon nanosequestration throughout the remediation process. Thus, the persistence of the nanocore remains an important unresolved question that is critical to understand for the implementation of such technologies.

Unfortunately, resolving the structure and dynamic interactions of assembled (co)polymers is impossible in many cases due to the lack of nanoprobes sensitive to molecular identity. In response, researchers turn to laborious super resolution fluorescent probes[19,25–27], transmission electron microscopy (TEM) staining[25,28], or small angle neutron scattering (SANS) with deuterium labeling[29,30], but such tagging can modify the phenomena of interest[31]. Even simply replacing $H_2O$ with $D_2O$ alters hydrogen bonding interactions and micelle structure significantly[32,33], impeding studies of bioactivity[34]. In addition, current neutron techniques require long data collection times (minutes to hours) that limit their measurement capability to only static samples. Optical light scattering can be used without labeling and can probe dynamic solutions but is resolution limited, while cryo-TEM techniques preclude measurements of dynamics and interactions[25,35,36]. Altogether, what is needed is an in situ nanoprobe with sensitivity to molecular identity that resolves the structure and dynamics of molecular assembly, evolution, and function of chemically distinct nanodomains.

Resonant soft X-ray scattering (RSoXS) combines the molecular bond sensitivity of near-edge X-ray absorption fine structure (NEXAFS) spectroscopy with the nanometer spatial sensitivity and statistical sampling of scattering. The technique has been used to identify electronic and magnetic spin states in topological materials[37,38] as well as measure polymer morphology and molecular orientation in solid-state organic nanostructures and devices[39–42]. In many of these cases, the unique chemical and electronic sensitivity of RSoXS has enabled measurements of (noncrystalline) ordering not possible with any other technique. Importantly, tuning the X-ray energy to a bond absorption resonance essentially labels the bond, similar to fluorescent tagging but without the need for disruptive chemical modification. Vacuum requirements of soft X-rays, however, impede measurement of liquid samples, especially at the carbon absorption edge—of strategic importance to organic materials. Thus, RSoXS has, to date, only been accomplished with hydrated pocket cells where success rate is low, non-repeatable sealing makes proper background data collection impossible, beam damage can be an issue, and rigorous quantitative analysis has not been developed[43,44].

We present here the construction and use of a liquid flow and mixing cell during RSoXS experiments enabling rapid data collection while solution conditions are changing. We utilize our microfluidic RSoXS instrument to measure a model drug delivery micelle system as proof of concept for this technique and demonstrate a 3-component spatiochemical analysis without the need for labeling. Such analyses have not been accomplished before by any technique on a single sample of aqueous polymer nanostructures. In addition, we investigate a polysoap nanocarrier to reveal that it uniquely exhibits a single-molecule structure that does not coalesce at high concentration—a characteristic critical to its application. This is the first use of X-ray scattering to probe polysoap nanostructure and dynamics. Even more, we characterize a single dynamic event occurring on the order of 30 s. Our label-free, in situ, and chemically quantitative RSoXS method has unlimited potential to probe dynamic solutions of stimuli-responsive and self-assembled systems, making it a significant advancement for the characterization of materials for biomedical and environmental applications.

## Results and discussion

**Creating label-free bond contrast in situ**. The scattering instrument is based on a TEM flow cell (shown in Fig. 1a, b) customized for insertion into the soft X-ray scattering endstation[45]. A commercial system was also employed for in situ TEM characterization. Both instruments flow a ~0.5-μm thick liquid channel between two 50 nm silicon nitride membranes to enable soft X-ray or electron beam penetration through the sample to an in-vacuum area detector. Dynamic mixing is possible via two inlet flow ports and one outlet port as pictured in Fig. 1a. This allows for solution conditions to change while the measurement is being taken.

We first demonstrate the technique on the Pluronic F127 micelle drug delivery platform[6,18], which has been chosen for its well-characterized structure and recent interest for biomedical applications like nano-scaffolding in live tissue bioprinting[46]. This provides for a means to compare our proof of concept results to already established literature observations. F127 is a triblock copolymer molecule with two hydrophilic polyethylene oxide (PEO) blocks separated by a hydrophobic polypropylene oxide (PPO) block (structure Fig. 1a inset). We investigate the micelle structure in both dry state (Fig. 1c) and aqueous (Fig. 1d) using the commercial TEM fluidic cell. Profiles of selected micelles represented in Fig. 1e show the hydrophilic micelle exterior extending into the surrounding water, but when dry, the structure collapses into a disc-like shape. This sensitivity of the nanostructure to its environment necessitates in situ studies. Although a careful survey could reveal the average micelle diameter, the micelles' internal chemical ordering cannot be ascertained, and it is uncertain whether these images are representative of the population.

Contrast in TEM is due to differences in electron densities. The failure of the TEM analyses to reveal the internal micelle structure

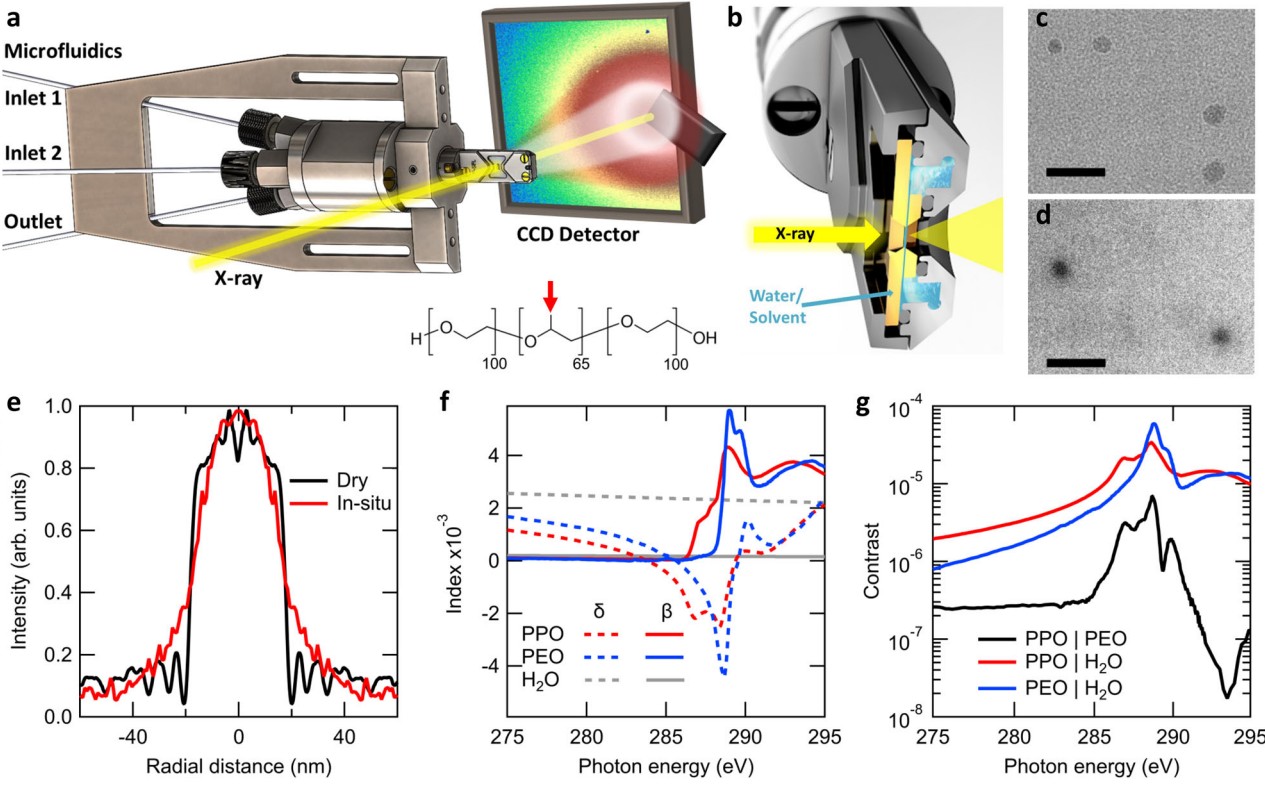

**Fig. 1 Microfluidic instrument enables multimodal characterization. a** Schematic of the instrument showing the microfluidic flow and mixing stage positioned with respect to the X-ray (electron) beam and CCD area detector containing a sample scattering pattern partially occluded by a protective beamstop. On the left, microfluidic lines (two in and one out) mix in the cell at the tip of the instrument. The metal fins to the left, top, and bottom enable mechanical mounting. Inset is the molecular structure of F127 with the unique methyl group of PPO indicated by the red arrow. **b** Cut away of the instrument tip showing the double silicon nitride membrane cell that enables liquid samples to flow through a 0.5-μm thick channel (determined by lithographically patterned spacers) vertically, while X-rays or electrons penetrate the channel horizontally. **c** Dry TEM images of F127 micelles acquired by allowing a ~0.1 wt. % aqueous solution to dry on a silicon nitride membrane. **d** TEM images of aqueous F127 micelles measured in solution under flow. Both image scale bars are 100 nm. **e** Cross-sectional profiles from the TEM-imaged micelles. Signal to noise is enhanced by averaging 180 radial profiles. **f** Optical constants Delta and Beta as a function of energy for PPO, PEO and water. **g** X-ray scattering contrast functions ($|\Delta n|^2$) between each of the components PPO, PEO, and $H_2O$, required to determine chemical composition of the nanostructure components.

originates from the similar electron densities of the two polymer blocks. RSoXS circumvents this issue via enhanced contrast at a molecular resonance specific to a unique chemical bond. The scattering intensity is proportional to the energy-dependent contrast function $I \propto |\Delta n(E)|^2$ where $\Delta n$ is the difference of index of refraction between two chemical moieties parameterized by real and imaginary components $n(E) = 1 - \delta(E) + i\beta(E)$. These components were determined at the carbon absorption edge for each polymer block via NEXAFS spectroscopy of pure PPO and PEO films (Fig. 1f). Near the edge (~290 eV), bond resonances can be seen in $\beta(E)$. In particular, the methyl group in PPO results in an extra absorption peak at 287 eV and reduced intensity of the backbone peaks at 289 eV. This enables RSoXS chemical contrast even without elemental differences. The contrast between each block (and water) is shown in Fig. 1g, with values continuously varying through four orders of magnitude. PPO contrast with water dominates below the edge and PEO contrast dominates at the resonance at 289 eV. This enables differential bond sensitivity varied continuously via the photon energy on one unmodified sample. Such intrinsic contrast variation in a nanoprobe is quite unique.

**Quantitative spatiochemical analysis.** Scattering patterns, shown in Fig. 2a, were acquired at photon energies in the range that maximizes the relative contrast between the three contrast functions shown in Fig. 1g and encompasses the desired variation

typically achieved by laborious chemical labeling. Two features appear in the patterns: a peak at $q \cong 0.22\ \text{nm}^{-1}$ and a shoulder at $q = 0.49\ \text{nm}^{-1}$ (see arrows). Notably, the peak position varies considerably with energy in the range $q = [0.19, 0.25]\ \text{nm}^{-1}$. The scattering signal from each feature also varied independently with energy, indicating that each has a chemically distinct source within the structure. Measurements were acquired both with and without flow occurring to confirm that shear forces within the cell did not affect the structure. It was found that using channel spacers >20× the nanoparticle size resulted in no difference in the scattering pattern with and without flow.

Although continuous contrast tuning in RSoXS has been demonstrated qualitatively, to date, quantitative spectral analysis has only been accomplished on integrated profiles that ignore the structure[42]. We developed a spectral scattering model to extract the full spatiochemical information of the nanoparticles. The model contains an energy-dependent, spherical, polydisperse core-shell form factor, $P_{cs}(\mathbf{q}, E)$, describing the chemically resolved internal structure (statistics of core and shell radii) and a Percus–Yevick hard-sphere structure factor, $S_{HS}(\mathbf{q})$, describing the packing behavior of the micelles (the closest approach and volume fraction occupied in solution). The core and shell are assumed to be homogeneous. Although the model's energy dependence based on a complex index is key to chemical sensitivity, the structural model is typical for scattering of micelle nanoparticles including Pluronics[30,47,48]. The Reduced 1-dimensional RSOXS

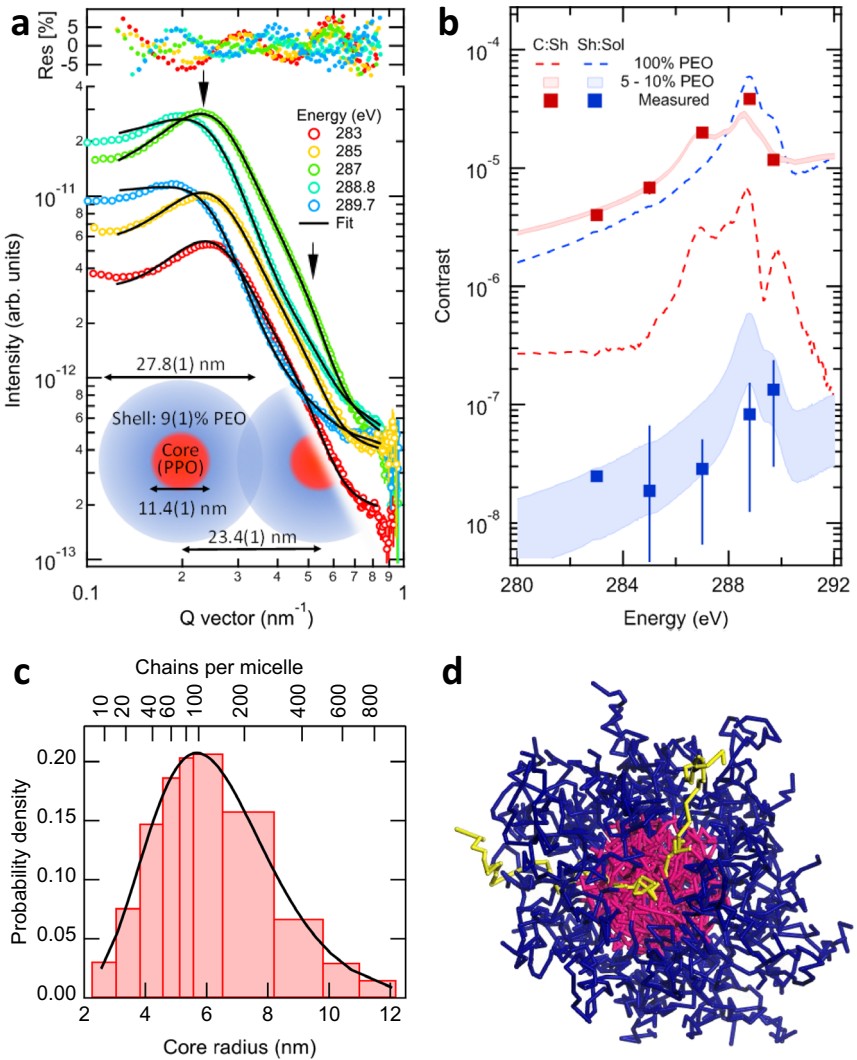

**Fig. 2 Structure and chemical composition analysis of aqueous F127 micelles. a** Simultaneous fit (black) of RSoXS profiles at selected photon energies revealing structural statistics. Uncertainties from counting statistics. Inset is schematic of the F127 micelle structure, molecular composition, and packing based on the fit results. See Methods for details. **b** Comparison of theoretical and measured contrast values from the structural fits for chemical analysis. C:Sh is the contrast between the core and shell of the micelle, while Sh:Sol is the contrast between the shell and solvent ($H_2O$). Uncertainties, represented by vertical lines, are described in detail in the Supplementary information. **c** Statistical distribution of core radii from fit results translated into number of aggregated chains within each particle, assuming a pure PPO core. **d** Rendering of the final micelle structures based on molecular dynamics simulations showing the pink PPO segments concentrated in the core while the blue PEO segments makeup a hydrated shell. Water molecules are not shown for clarity. The triblock nature of the molecules (PEO-PPO-PEO) is demonstrated by highlighting a single molecule as yellow.

profiles were fit to Eq. (1):

$$I(\mathbf{q}, E) = A \cdot T(E) \cdot P_{cs}(\mathbf{q}, E) \cdot S_{HS}(\mathbf{q}) + I_{bkg}(\mathbf{q}, E) \qquad (1)$$

where $\mathbf{q}$ is the momentum transfer vector, $E$ is the photon energy, $A$ is a global scale parameter, $T(E)$ is an attenuation parameter, and $I_{bkg}(\mathbf{q}, E)$ is an energy-dependent background. Details of these functions and fitting procedure can be found in the Supplementary Methods. This model was simultaneously fit to all scattering profiles shown in Fig. 2a, with a schematic representation of the average extracted structure parameters inset (all physical parameters in Supplementary Table 2, separate Structure Factor and Form Factor contributions shown in Supplementary Fig. 12). These parameters could not be resolved by fitting any individual scattering profile but required a simultaneous fit for success.

The statistical distribution of core radii, exhibited in Fig. 2c, reveals that 20–400 chains coalesce into one micelle (assuming

only PPO in the core). The mode of this distribution has a micelle diameter of 27.8(1) nm, representing the most probable size of micelle in the distribution, which is in good agreement with previous reports[36,48,49], and indicates the average micelle contains 90 polymer chains. Noteworthy is that the closest approach between micelles is less than their diameter, which indicates an interpenetration of adjacent micelles, as has been previously hypothesized[50]. A geometrical argument explains why this is possible. The (pure PPO) core radius and polymer block molecular weights determine the quantity of PEO material in the shell. The shell radius is quite large, however, requiring ~88% $H_2O$ to balance the volume. This shell hydration enables nanoparticle interpenetration.

This analysis not only reveals the internal structure and packing of the micelle without the need for labeling but also the quantitative molecular composition of that structure. Determination of the $H_2O$ content of the shell is also determined

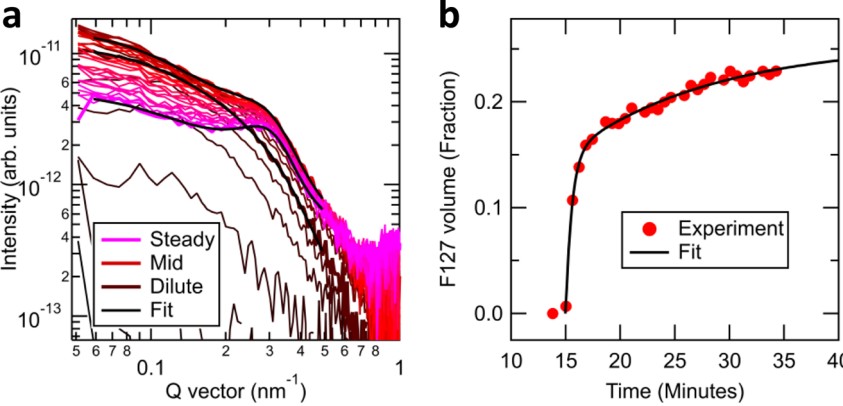

**Fig. 3 Aqueous F127 micelle dynamics. a** In situ RSoXS profile at 287 eV of 1 wt. % F127 replacing DI H$_2$O within the flow cell. Fits of selected scattering profiles overlaid involving a spherical form factor and hard-sphere structure factor. See Methods for details. The color of the trace represents 30 s time intervals from brown (dilute) through red (mid purge) to pink (steady state). **b** Micelle solution volume fraction versus time from profile fits in (**a**). A double exponential fit is overlaid starting at 15 min into the transition.

independent of the feature sizes extracted from the fit. We accomplish this through contrast function analysis. For each profile that is fit, a contrast $|\Delta n|^2$ between each of the three components of the nanostructure (core, shell, and solvent regions) is extracted as a fit parameter for each photon energy from within $P_{cs}(\mathbf{q}, E)$. From our optical constants in Fig. 3f, we calculate the expected contrast functions. A chemically mixed region of the nanostructure is represented by a linear combination of optical constants (complex index) weighted by their respective concentrations. These weights become fit parameters of the chemical composition within the nanostructure components. Details of the fitting procedures used in this analysis can be found in the Supplementary Methods.

Figure 2b shows the results of this quantitative chemical analysis to the extracted contrast values (symbols) and two example scenarios of the chemical composition within the nanostructure. The results of the analysis indicate a pure PPO core (red region, <5% water) and a hydrated PEO shell (blue region) with concentration of 9(1)% PEO in water, agreeing well with the crude geometric estimate discussed above. For comparison, contrast functions for a pure PEO shell (dashed) are qualitatively different from the extracted values, and the relative magnitudes are reversed. This indicates a high measurement sensitivity for RSoXS to chemical composition. The scenario of a PEO core and PPO shell can also be definitively eliminated by this analysis. The sensitivity of this analysis is remarkable given that there is only a single unique methyl bond distinguishing PPO from PEO.

The power of this analysis lies in the ability to quantify any number of $n$ unique chemical moieties within a structure, which only requires recording the scattering pattern at $n$ photon energies, strategically chosen based on the measured bond resonances. A similar chemical analysis has been accomplished using SANS with variable deuteration of the polymers to vary the contrast. However, this requires multiple, uniquely labeled samples for three or more chemical components. Deuteration is both chemically laborious and has been shown to alter micelle structure and dynamics[31–33], degrading results. The instrument and analysis presented in this work uniquely enables the complete spatiochemical analysis on one, unmodified sample in situ.

We modeled the micelle with dissipative particle dynamics (DPD) simulations starting with a random solution of the polymers in water with an example rendering displayed in Fig. 2e. A composition analysis of the simulated structure confirms a water-free PPO core and a hydrated PEO corona with 78% H$_2$O.

Although the model is not quantitatively accurate, it supports our results of a highly hydrated shell. Additional information on the DPD simulation is located in the Supplementary Methods. The extreme extent of the shell hydration that we measure here explains why these materials are biocompatible and stable under biological conditions, allowing for their use as drug delivery vehicles. Such a capability for RSoXS to clearly and quantitatively measure the composition based on a single unique bond moiety will be powerful for investigations of drug loading, release rate, and formulation stability of new smart medicine delivery systems.

**Operando micelle dynamics.** Although characterizing polymer nanostructure and molecular composition in situ is powerful, development of functional nanomaterials often requires monitoring interactions dynamically. The dual flow design of the instrument allows for continuous mixing and reaction of samples, enabling titration and other concentration sensitive experiments. Flow also greatly mitigates beam interaction and sample degradation effects often associated with intense nanoprobes. Replacing one sample with another is perhaps the simplest example of dynamics and is demonstrated in Fig. 3a by an abrupt step-function transition from H$_2$O flow to that of the F127 solution. The transition was monitored via sequential CCD exposures at a constant scattering photon energy. At early times in the transition, the scattering increases and develops into an isolated spherical form factor (dilute). With time, a structure factor peak emerges, originating from scattering between micelles (steady) due to the increased micelle-micelle interaction with concentration. Remarkably, less than 50 μL of liquid (0.5 μg dry material) flowed through the cell for the entire experiment, rendering the technique amenable to bioassays.

We accomplished a similar spatiochemical analysis as above with only the volume fraction and overall intensity parameters allowed to vary throughout the process. The resulting volume fraction dynamics are shown in Fig. 3b, which in turn fits well to a double exponential dynamical model with time constants $\tau_1 = 34(3)$ sec and $\tau_2 = 13(4)$ min. The latter time is consistent with the onset of dynamics $t_0 = 15$ min, which is the lag time of the microfluidics system based on tubing length and flow rate. The shorter time is interpreted as the dynamics of fluid flow through the cell rather than the dynamics of micelle assembly, which has been previously characterized as 100 ms or faster[51]. Optimizing cell area to the significantly larger X-ray beam and upgrading to the planned higher sensitivity detector is expected to enable such time resolution, which is also on the order of polymer aggregation

and crystallization processes[52]. It is notable that the in situ nanoscale dynamics measured here on unlabeled polymer micelles is not possible with any other technique. In particular, this time resolution is not possible with SANS. In addition, concerns arising from fluorescent tagging[8,21,53], nanoparticle labeling[54,55], or even deuteration[31–33] that alter micelle structure and dynamics are eliminated via RSoXS.

**Unimeric cores within aggregates for hydrocarbon nanosequestration**. We now turn to the question of polysoap internal structure of interest for oil spill remediation and drug delivery applications[10,11,24]. A recent polysoap, Poly(AMPS-stat-DDAM), a statistical copolymer of 2-acrylamido-2-methylpropane sulfonic acid (AMPS) and n-dodecyl acrylamide (DDAM) (aPS50, molecular structure Fig. 4b inset), has shown promise with uptake of hydrocarbon molecules within the hydrophobic DDAM-rich core domains[11]. We confirm hydrocarbon uptake in aPS50 using pyrene as a spectroscopic model compound (see Supplementary Methods and Supplementary Figs. 5 and 6). Both individual unimeric micelles and large aggregates have been observed[11,23,56], but only limited information about the assembled structure has been obtained. For example, light scattering and viscosity techniques have been used to directly probe the overall size of structures, and spectrophotometric probes have been used to indirectly determine relative sizes and aggregation numbers of microdomains using fluorescent labels[57]. No study has been successful in resolving core dynamics and structure. Whether aggregates retain unimeric cores or the cores coalesce into multimers (as in the case of F127 shown above) has been a longstanding question, and this distinction is critical to their performance, since core coalescence drastically reduces surface area for hydrocarbon capture. Figure 4a shows dry TEM images of aPS50 particles at low concentrations revealing ~20 nm diameters. However, aqueous TEM measurements of the micelles were unsuccessful. Our own dynamic light scattering (DLS) studies on aPS50 reveals an increase in particle size with concentration (Fig. 4b), ranging from a hydrodynamic radius of 36 nm at low concentrations (0.1 g/L) to 98 nm at 10 g/L, consistent with mesoscale aggregates. However, information on core coalescence cannot be resolved.

Here, we present the first example of using an X-ray scattering technique to characterize a polysoap system with the spatial resolution to probe core structure and dynamics that have not been observed prior. RSoXS scattering profiles of the dynamic flow transition between $H_2O$ and an aPS50 solution are shown in Fig. 4c. During the transition, the scattering profiles are consistent with that of static concentration controls (see Supplementary Fig. 8), indicating a quasi-equilibrium state of the structure during the transition. In contrast to the F127 behavior, the aPS50 scattering profiles exhibit a structure factor feature even at dilute concentrations, strengthening and shifting to higher $q$ values as concentration increases. This plus the high intensity at low $q$ indicates a high propensity of the particles to form aggregates. Analysis (Fig. 4d) reveals that throughout the entire concentration range, the form factor provides a median radius of 2.8(4) nm, while the structure factor radius (i.e., particle spacing) decreases dramatically from 22 nm under dilute conditions (0.1 wt. %) to 10 nm at 2 wt. %. The hydration of the structures results in a nearly 30% volume fraction at maximum concentration, a value near that of a hydrogel structure. It is of note that through flow, the entire concentration series was measured within 5 min, making such measurements capable of high-throughput characterization.

Our results of a decreasing particle distance but constant form factor radii indicate an increasing proximity of unchanging cores

as concentration increases. The light scattering data (Fig. 4b) and strengthening RSoXS structure factor (Fig. 4d) indicate that these micelles, unimeric under dilute conditions, are aggregated within ever larger clusters as concentration increases. However, the unchanging core size with concentration demonstrates that the cores remain unimeric within the aggregates, forming a hierarchical structure consistent with the model of flower-like micelles[56]. This model, depicted in Fig. 4e, exhibits an open structure with bridging chains that are outstretched at low concentrations due to electrosteric repulsion between the charged coronas[11,23,56]. These bridges compress during aggregation but allow massive interfacial area of the cores for hydrocarbon access and capture that is not available in structures such as F127. These measurements provide the first direct experimental support for unimeric micelle bridging rather than coalescence of micelle core domains upon aggregation. Our results further determine that such a structure persists from dilute to saturated volume fractions, even when the micelles associate into larger multi-micelle structures, which is important for their utility as organic nanocarriers.

To date, the chemical and electron spin sensitivity of RSoXS has proven powerful in solid-state materials. However, our application to aqueous media paired with a spatiochemical analysis now quantifies structure, composition, and dynamics of polymeric or biological assemblies that is impossible in any other way. With the elimination of labeling requirements and sensitivity to even a single unique bond within a molecule, characterization and engineering of molecular nanostructure evolution and interactions is now much more accessible. Specifically, in the case of polysoap micelle structures, the revelation of a stable unimeric core from dilute to high concentrations further validates their potential for applications, such as targeted drug delivery and environmental/water remediation. The implications of being able to measure dynamic solutions in situ and label-free expands to the study of pH-, salt-, and analyte-responsive polymers and self-assembled materials that are particularly important in biomedical and environmental applications.

## Methods

**Materials and sample preparation**. Pluronic F127, or just F127 for shorthand, was purchased from Sigma Aldrich, and used as received. F127 is a block copolymer of 100 ethylene oxide (EO) units followed by 65 propylene oxide (PO) units, followed by 100 EO units, with an average molecular weight of 12.6 kDa. PPO and PEO were purchased from Sigma Aldrich. AMPS (50% v/v in water), 2-cyano-2-propyl dodecyl trithiocarbonate (CPDT) (>97%), and azobisisobutyronitrile (AIBN) (98%) were purchased from Sigma Aldrich. DDAM (>97%) was purchased from TCI America and used as received. The monomer, AMPS, was isolated and purified via precipitation from acetone and collected via vacuum filtration and dried under high vacuum. The initiator, AIBN, was purified by recrystallization from methanol. The RAFT chain transfer agent, CPDT, was purified via column chromatography prior to use.

The polysoap chosen for this study was synthesized according to previous procedures utilizing the statistical RAFT copolymerization of equal molar ratios of AMPS and DDAM[11]. A brief description of the synthesis is included in the Supplementary Methods. Structural data for the aPS50 polymer were collected using proton nuclear magnetic resonance ($^1H$-NMR) spectroscopy (Supplementary Fig. 10) for determining %DDAM content and gel permeation chromatography (GPC) (Supplementary Fig. 11) for molecular weight and PDI information. $^1H$-NMR measurements were collected using Varian MercuryPLUS 300 MHz NMR spectrometer in deuterated methanol with a delay time of 5 s. GPC measurements were performed with a Viscotek TDA302 triple detector array of RI, low- and right-angle light scattering, and viscosity detectors. The GPC system was equipped with TOSOH Biosciences TSK-Gel columns (SuperAW3000 and SuperAW4000). The eluent was 0.2 M $LiClO_4$ in methanol at a flow rate of 0.6 mL/min.

**Transmission electron microscopy**. TEM data were collected at the National Center for Electron Microscopy, part of the Molecular Foundry at Lawrence Berkeley National Laboratory (LBNL). Bright field TEM images were collected on an image-corrected Themis microscope with a high brightness XFEG source, operated at 300 kV, and imaged on a high-speed Ceta camera.

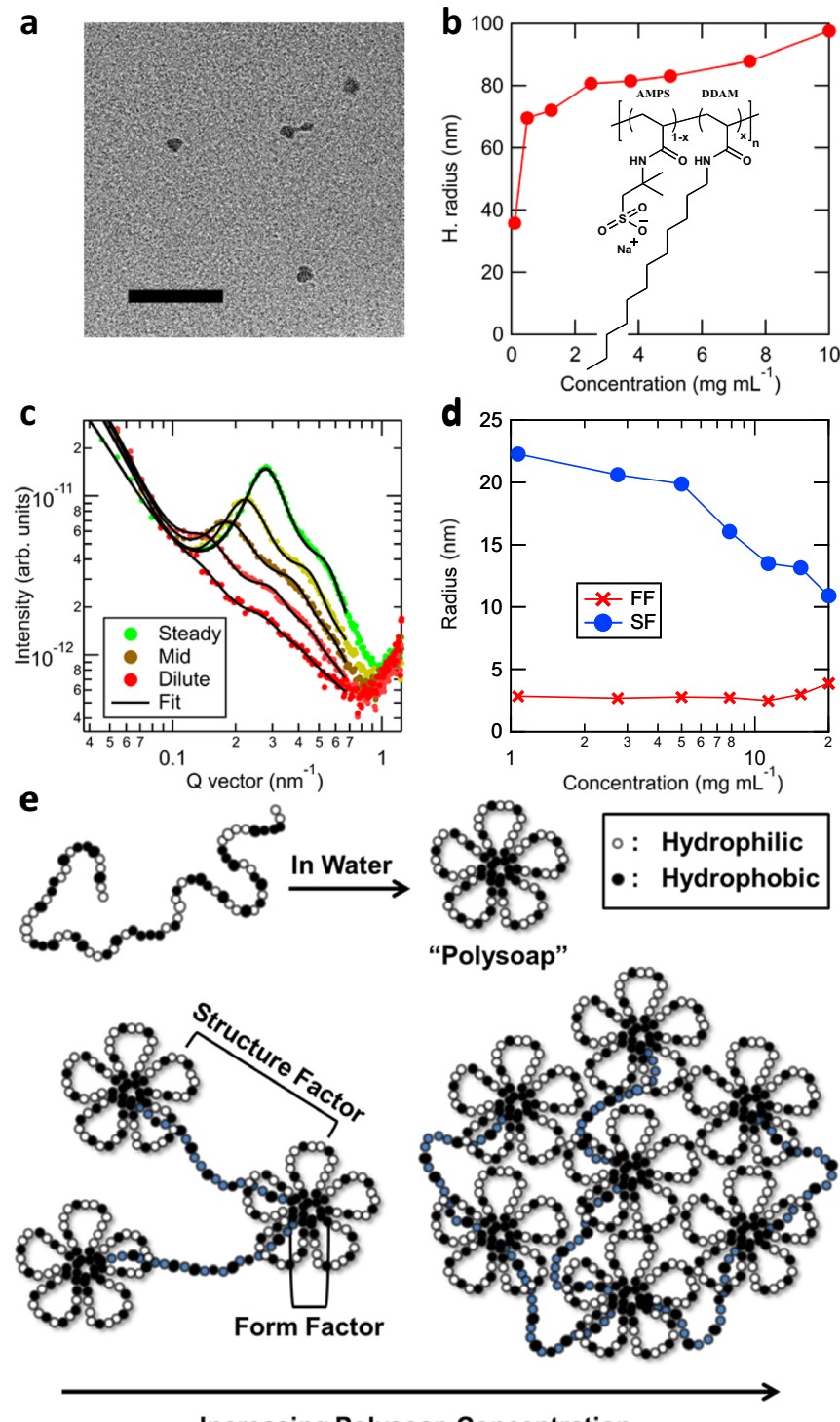

**Fig. 4 Multimodal investigation of aPS50 micelles. a** Dry TEM images acquired by allowing a 2 wt. % aqueous aPS50 solution to dry within the instrument. Scale bar is 100 nm. **b** Micelle hydrodynamic radius as a function of concentration of an aPS50 solution in water acquired by light scattering. aPS50 Molecular structure is inset, where $x = 0.5$. **c** Selected in situ RSoXS profiles at a photon energy of 287.3 eV of 2 wt. % aPS50 replacing DI $H_2O$ within the flow cell. Fits of scattering profiles represent a spherical form factor with a hard-sphere structure factor, see Supplementary Fig. 13 for individual Structure Factor and Form Factor contributions. **d** Fit results from (**c**) displaying the structure factor and form factor radii vs concentration, calibrated from dynamic flow time using static concentration controls. **e** Illustration of the self-assembly of aPS50 based on the results. The blue shaded chains highlight bridging tie chains between micelles.

The Dry F127 image shown in Fig. 1c was collected on a sample prepared by drop-casting a ~0.1 wt. % solution onto a 50 nm silicon nitride window and allowing it to dry in air. In situ F127 TEM was measured using a Poseidon Protochips TEM holder with two crossed $50 \times 550$ μm windows, 50 nm thick; 500 nm flow spacer was used. In situ TEM data on aPS50 were attempted, but we could not resolve any micelles nor aggregates. Dry aPS50 micelles shown in Fig. 4a were imaged by using the electron beam to drive the water rapidly out of the field of view. The intensity vs radial distance plot in Fig. 1e was calculated by azimuthal averaging of selected example particles from Fig. 1c, d. Consequently, the data at the center of the cell remain rather noisy.

**Liquid RSoXS instrument and experimental procedures**. RSoXS was performed at BL 11.0.1.2[45] of the Advanced Light Source (ALS) at LBNL using a custom Protochips liquid flow cell. A back-illuminated Princeton PI-MTE CCD cooled to −45 °C detected the scattering pattern with exposures between 30 and 120 s. One drawback of soft X-rays is there need for high vacuum, making measurement of liquids more challenging. We overcome this challenge by using a custom flow cell created by Protochips based on their in situ TEM products and design help from the authors. It utilizes a silicon nitride membrane window, supported by a silicon wafer frame, to provide a barrier between the liquid sample and the vacuum chamber, while allowing X-rays to be transmitted. Windows were purchased from Protochips chips with $50 \times 550$ μm wide windows, which were crossed to provide a $50 \times 50$ μm transmission window—significantly smaller than the 250 μm X-ray beam diameter. A 500 nm flow spacer was chosen to avoid confinement effects when flowing the ~25 nm micelles, while also prevented unnecessary attenuation of the X-rays from the increased path length through the sample. Syringe pumps are used to provide reliable flow rates in the range of 25–600 μL/h, which can be all through one inlet, or distributed between both inlets.

Since this technique is developed for online monitoring of structural changes during the experiment, both inlet feed lines are used. Specifically, in our initial set of experiments, one inlet is for de-ionized (DI) water ($H_2O$), and the other is for a solution of F127 (1% by mass in DI $H_2O$). To start with, the sample cell is loaded with DI $H_2O$ to allow for background scattering measurements to be acquired. No difference in scattering profile was observed between the static and flow measurements of DI $H_2O$. Also, all X-ray data shown in this manuscript are from sample measurements under flow, which has the added benefit of preventing possible X-ray beam effects, such as damaging the chemical bonds in the polymer and causing sample degradation, as well as making sure the sample is homogeneous. In order to generate a proper background, a large range of scattering energies was measured while DI $H_2O$ was flowing. Once a stable background is established in DI $H_2O$, the solution flow is fully switched to the F127 solution feed to establish a baseline scattering profile for 100% of the sample solution. During actual data collection the flow from each inlet feed can vary, 100% DI $H_2O$ to 100% sample solution, thus giving us a range of solution conditions that can be probed online. Additional discussion of sample integrity considerations is in the Supplementary Discussion.

**RSoXS data processing**. Raw RSoXS scattering patterns collected on the 2D area detector were reduced to 1D scattering profiles via azimuthal averaging in the NIKA processing package developed by Jan Ilavsky with a modified handling of the energy dimension via a custom panel[58] and processing functions in IGOR Pro 8 (Wavemetrics). This included standard processing for experimental geometry, corrections for energy-dependent incident beam intensity, and dark background subtraction.

All RSOXS model analysis was computed with the IRENA Modeling II package developed by Jan Ilavsky[59] with custom script modifications that allow for simultaneous multi-energy fits. The built-in Levenberg–Marquardt algorithm in the IGOR Pro 8 software environment was used for all fitting.

**Optical constants**. PPO was drop-cast from the as-received solution, and excess solution was wicked off using a wipe. Due to the molecular weight of the PPO, it remained as a liquid with a very slow evaporation rate, allowing it to be loaded into the high vacuum chamber as a liquid. PEO was dissolved in $H_2O$ prior to being spun-cast as a dry film. Total electron yield (TEY) NEXAFS spectrum of PEO was collected at BL 6.3.2[60], and TEY NEXAFS of PPO was collected at BL 11.0.1.2[45] of the ALS at LBNL. The measured fine structure was scaled to bare atom mass absorption calculated from the CXRO database, enabling calculation of the imaginary component of the index of refraction Beta. Beta for $H_2O$ was taken directly from the CXRO database. The real component of the index of refraction (Delta) was calculated from Beta using Kramers–Kronig transformation methods[61]. Since water interactions with PPO or PEO involve only weak Vander Waals interactions, the optical constants measured in this way are expected to be identical to those of the polymer when dissolved in water.

**Concentration dynamics**. To monitor the dynamics of micelles replacing pure water in the cell, repeated RSoXS scans were collected, as shown in Fig. 3a. A flow rate of 75 μL/h was chosen to slow down the transition to an appropriate timescale for our desired scan given that acquisition times of 30 s were needed for good signal to noise. The scattering profiles were background corrected by subtracting the data collected from the cell before flowing the micelle solution, when it was pure $H_2O$. Additional details are in the Supplementary information.

**Dynamic light scattering**. DLS measurements were collected using incident light at 633 nm from a Research Electro Optics HeNe laser operating at 40 mW. The time-dependent scattering intensities were measured from a Brookhaven Instruments BI-200SM goniometer at 60, 75, 90, 105, and 120 degrees with an avalanche photodiode detector and TurboCorr correlator.

**Characterization of aPS50 in water via UV-Vis and fluorescence spectroscopy**. UV-Vis spectroscopy and fluorescence spectroscopy for pyrene absorbance

and fluorescence were measured with a TECAN Safire 96-well plate spectrometer running on integrated Microsoft Excel software. Absorbance was measured at 341 nm, as shown in Supplementary Fig. 5, and fluorescence was measured via emission scan from 350 to 550 nm with an excitation wavelength of 341 nm.

## Data availability
The data that support the findings of this study are available from the corresponding author upon reasonable request.

## Code availability
The code that is used and developed in this study is available from the corresponding author upon reasonable request.

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

## Acknowledgements

Support for T. M. was provided by the National Science Foundation Major Research Instrumentation grant 1626566. Support for T. F. and B. A. C. was from the Department of Energy Early Career Research Program under grant DE-SC0017923. I. A. C. was supported through LDRD and Advanced Light Source grants through the US Department of Energy, Office of Science, Basic Energy Sciences, Scientific User Facilities Division under contract number DE-AC02-05CH11231. P. D. P. was supported by the National Science Foundation's Experimental Program to Stimulate Competitive Research (EPSCoR) under Cooperative Agreement No. IIA1430364, by the CMEDS Consortium of GOMRI (grant# SA 12-05/GoMRI-002), and by the National Science Foundation Graduate Research Fellowship Program under Grant GM004636/GR04355. This research used resources described above of the Advanced Light Source, which is a DOE Office of Science User Facility under Contract No. DE-AC02-05CH11231. Work at the Molecular Foundry was supported by the Office of Science, Office of Basic Energy Sciences, of the US Department of Energy under Contract No. DE-AC02-05CH11231.

## Author contributions

B. A. C. conceived and oversaw the project with major input from T. M. T. M. carried out the project, including RSoXS and TEM measurements and analysis as well as MD DPD simulations. T. F. developed and conducted the spatiochemical quantitative analysis. C. W. and I. A. C. designed and developed the custom electrochemical flow RSoXS instrument. C. L. M. and P. D. P. synthesized the aPS50 polysoap molecules and conducted the light scattering, pyrene absorbance, and pyrene fluorescence measurements and analysis. The manuscript was written by T. M. and B. A. C. with major contributions from P. D. P. and some contributions from C. W., I. A. C., and T. F. as well.

## Competing interests

The authors declare no competing interests.
