## [Peer Review File · Nature Communications]

REVIEWER COMMENTS

Reviewer #1 (Remarks to the Author):

In this study, the authors show a method to elucidate the spatial distribution of components inside polymer micelles without labeling using the microfluidic RSoXS instrument newly constructed by the authors, and further, they apply this method to characterize a new type of polysoap DDS carrier. The RSoXS method, which has been developed by Wang et al, can visualize the spatial distribution of light elements, such as C, N, O, without even deuteration labeling. So, the RSoXS is an extremely useful for structural analyses of soft materials in bulk. Although the reviewers have considered that it difficult to apply RSoXS to aqueous solutions of polymer micelles, the authors have shown that it is possible with acceptable accuracy by using their RSoXS instrument. This result is evaluated as having an extremely high impact in the structural analysis of soft materials in solutions. Further, the application of this system to elucidating the association behavior of new polysoap shows that microfluidic RSoXS will be a powerful tool in DDS characterization. The measurement method and data analysis procedures are rational and can be evaluated as sufficiently reliable results. Therefore, the reviewer considers this article to be worth publishing in Nature Communication.

Reviewer #2 (Remarks to the Author):

The article reports on a new implementation of a X-ray scattering technique using resonant soft X-rays (RSoXS). The resonant nature of the scattering process allows distinguishing different moieties in a nanoscale structured material. In the present work, this method is applied to the well studied pluronic micelles and a more novel and less studied polysoap structure aPS50. The RSoXS technique is known in principle and has been validated. However, due to vacuum requirements, one could not apply it routinely to aqueous solutions in the past. In their work, the authors overcome this challenge by designing a microfluidic flow and mixing cell, where the latter also allows 'rapid data collection while solution conditions are changing'. The authors also (correctly) state that solution-RSoXS has to be seen compared to other scattering techniques on solutions, notably small-angle neutron scattering. The advantage of their solution (RSoXS in a microfluidic cell) is that there is no need for chemical modifications/labeling of the different entities. In my view, the challenges are the use of the microfluidic cell and the fact that the materials x-ray resonances have to be present and accessible for specific material composition of choice. Future studies will show whether this represents a major obstacle towards a widespread use of the technique. Overall I find the article very interesting, well written, and convincing. If RSoXS of polymers and surfactant systems in solutions could replace a substantial part of neutron scattering experiments, this would significantly advance the field. I, therefore, recommend the publication of the article in its present form.

Reviewer #3 (Remarks to the Author):

The manuscript by McAfee et al reports on the use of RSOXS to study in solution polymeric micelles. The study demonstrates a contrast variation capability without labelling that is very interesting and of dramatic potential application in different fields of polymer science.

The study shows data on a classic core-shell system such as pluronic and a novel poly soap compound.

The manuscript is well written and dense of important findings, showing the potentiality of the technique but also reporting interesting scientific findings.

However, before publication in Nat Comm, I have some questions reported below. Particularly questions 6 to 11 are very important and I urge the authors to provide comments and answers.

1) Line 87-88: The sentence “The scattering instrument is based on a TEM flow cell (shown in Figure 1a and b) customized for insertion into the soft X-ray scattering instrument” needs some revision. Confusion between the scattering instrument and the soft scattering instrument.

2) Line 88: “~0.5 μm ” use the correct symbol for micrometer.

3) Line 117: How can the authors state that F127 collapsed in a well defined disk? Do they know the thickness of the dry micelle?

4) Line 133: “on one unmodified sample” This statement actually made me thinking about flow. Is the sample measured under flow? If so, can the authors be sure that flow does not influence the results?

If no flow is applied, have the authors test for possible radiation damage that it is not negligible a-priori for these soft resonant energies?

5) Line 175: Considering the polydispersity of the micelles, stating 91 chains in average is too ambitious. Just stick to 90.

6) Line 175-177: Here, interpenetration of the micelles is claimed. I could not fully understand what is the concentration of F127 for the data reported in Figure 2. Is 1% w/w mentioned in the SI? If so, I am surprised that the authors find a structure factor contribution that strong and, even more strange, that they see interpenetration.

What is the volume fraction as calculated from the PY fit? Please comment on this point.

7) Line 182: “quantitative molecular composition of that structure, independent of the geometry”.

What do the authors want to say here?

They have actually used a geometrical model for the spherical objects. Please comment.

Also, what is the error in the contrast number that they obtain from the fit?

8) Line 138-139: the authors mention that the peak position shifts with changing the energy. If this is coming from a structure factor, shouldn't this be energy independent? Indeed they write $S(q)$ and not $S(q,E)$.

So, why the peak moves?

9) Have the author tested that the poly soap (and the F127 as well) do not stick to the wall and cumulate during flow?

10) The part dealing with the pyrene uptake is not clearly presented. Is the poly soap as investigated by RSOXS loaded with pyrene or not?

If not, why reporting all the pyrene loading data?

If yes, please state this clearly and also what loading, etc.

11) in order to judge the quality and validity of the fit, I require the authors to present in the SI for some selected energies all the fitted components (structure and form factor independently).

For the F127, but especially for the poly soap, where the unchanging core dimension is claimed

In summary, I believe that the manuscript reports a relevant study that is potential for publication in Nature Communication after the points raised by the reviewer are addressed.

Regards

Giuseppe Portale

Greetings Reviewers,

We are elated with the reviewer responses and thank them for their work. We are very pleased that all of the reviewers take note of the great potential this novel technique has to impact research on soft materials in solution. Please find below our **point-by-point response (red text)** to each of the **reviewer #3's comments (Blue text)**.

Reviewer #1 (Remarks to the Author):

In this study, the authors show a method to elucidate the spatial distribution of components inside polymer micelles without labeling using the microfluidic RSoXS instrument newly constructed by the authors, and further, they apply this method to characterize a new type of polysoap DDS carrier. The RSoXS method, which has been developed by Wang et al, can visualize the spatial distribution of light elements, such as C, N, O, without even deuteration labeling. So, the RSoXS is an extremely useful for structural analyses of soft materials in bulk. Although the reviewers have considered that it difficult to apply RSoXS to aqueous solutions of polymer micelles, the authors have shown that it is possible with acceptable accuracy by using their RSoXS instrument. This result is evaluated as having an extremely high impact in the structural analysis of soft materials in solutions. Further, the application of this system to elucidating the association behavior of new polysoap shows that microfluidic RSoXS will be a powerful tool in DDS characterization. The measurement method and data analysis procedures are rational and can be evaluated as sufficiently reliable results. Therefore, the reviewer considers this article to be worth publishing in Nature Communication. **We thank Reviewer 1 for this positive assessment and in particular agree that "This result is evaluated as having an extremely high impact in the structural analysis of soft materials in solutions"**

Reviewer #2 (Remarks to the Author):

The article reports on a new implementation of a X-ray scattering technique using resonant soft X-rays (RSoXS). The resonant nature of the scattering process allows distinguishing different moieties in a nanoscale structured material. In the present work, this method is applied to the well studied pluronic micelles and a more novel and less studied polysoap structure aPS50. The RSoXS technique is known in principle and has been validated. However, due to vacuum requirements, one could not apply it routinely to aqueous solutions in the past. In their work, the authors overcome this challenge by designing a microfluidic flow and mixing cell, where the latter also allows 'rapid data collection while solution conditions are changing'. The authors also (correctly) state that solution-RSoXS has to be seen compared to other scattering techniques on solutions, notably small-angle neutron scattering. The advantage of their solution (RSoXS in a microfluidic cell) is that there is no need for chemical modifications/labeling of the different entities. In my view, the challenges are the use of the microfluidic cell and the fact that the materials x-ray resonances have to be present

and accessible for specific material composition of choice. Future studies will show whether this represents a major obstacle towards a widespread use of the technique. Overall I find the article very interesting, well written, and convincing. If RSoXS of polymers and surfactant systems in solutions could replace a substantial part of neutron scattering experiments, this would significantly advance the field. I, therefore, recommend the publication of the article in its present form.

We thank Reviewer 2 for this positive review and in particular agree that “If RSoXS of polymers and surfactant systems in solutions could replace a substantial part of neutron scattering experiments, this would significantly advance the field.”

While we are humbled that both reviewer #1 and #2 recommend publishing the article without any revisions, we are especially grateful for reviewer #3’s detailed constructive comments and questions.

All of reviewer #3’s comments shed light on areas of the manuscript that needed some additional information or clarification, and addressing those comments has made the manuscript even stronger. We addressed all of reviewer #3’s comments with modifications to either the main text, the SI, or both. In response to Reviewer #3’s questions 4), 6), and 9), a new section titled “Sample Integrity Considerations” was added to the SI so that all information concerning the sample integrity was in one easy-to-find place.

Reviewer #3 (Remarks to the Author):

The manuscript by McAfee et al reports on the use of RSOXS to study in solution polymeric micelles. The study demonstrates a contest variation capability without labelling that is very interesting and of dramatic potential application in different fields of polymer science. The study shows data on a classic core-shell system such as plutonic and a novel poly soap compound. The manuscript is well written and dense of important findings, showing the potentiality of the technique but also reporting interesting scientific findings. However, before publication in Nat Comm, I have some questions reported below. Particularly questions 6 to 11 are very important and I urge the authors to provide comments and answers.

We thank the reviewer for these positive comments indicating that he is in favor of publication in Nature Communications. Below are our point-by-point response and description of manuscript changes to his excellent questions and suggestions. We accept all of his suggestions.

1) Line 87-88: The sentence “The scattering instrument is based on a TEM flow cell (shown in Figure 1a and b) customized for insertion into the soft X-ray scattering instrument” needs some revision.

Confusion between the scattering instrument and the soft scattering instrument.

The reviewer is correct that we used “scattering instrument” to mean two different things in the same sentence. The text has been altered to correct this.

2) Line 88: “~0.5 um“ use the correct symbol for micrometer.

This has been corrected.

3) Line 117: How can the authors state that F127 collapsed in a well defined disk? To they know the thickness of the dry micelle?

The TEM data is not suitable for quantitative thickness determination. The claim in the text has been changed from “well defined disk” to “disk-like shape”.

4) Line 133: “on one unmodified sample” This statement actually made me thinking about flow. Is the sample measured under flow? If so, can the authors be sure that flow does not influence the results?

If no flow is applied, have the authors test for possible radiation damage that it is not negligible a-priori for these soft resonant energies?

By unmodified, we mean that invasive labeling, such as deuteration, is not required. We continuously flow the sample to avoid beam damage. We performed experiments ensure the scattering profile when flowing the sample is the same as static conditions.

We added a section to the SI called “Sample Integrity Considerations“, which discusses these important question.

Additionally, we note that we do indeed state in the manuscript on line ~141 “It was found that using channel spacers >20x the nanoparticle size resulted in no difference in the scattering pattern with and without flow. “

5) Line 175: Considering the polydispersity of the micelles, stating 91 chains in average is too ambitious. Just stick to 90.

The reviewer has a good point. The text has been changed to state “the average micelle contains 90 polymer chains”.

6) Line 175-177: Here, interpenetration of the micelles is claimed. I could not fully understand what is the concentration of F127 for the data reported in Figure 2. Is 1% w/w mentioned in the SI?

If so, I am surprised that the authors find a structure factor contribution that strong and, even more strange, that they see interpenetration.

What is the volume fraction as calculated from the PY fit? Please comment on this point.

Figure 3 shows F127 volume fractions larger than 20% for 1 wt. % F127, which indeed deserves some additional consideration and explanation. This has been addressed in the new section added to the SI called “**Sample Integrity Considerations**”.

7) Line 182: “quantitative molecular composition of that structure, independent of the geometry”. What do the authors want to say here?

They have actually used a geometrical model for the spherical objects. Please comment.

Also, what is the error in the contrast number that they obtain from the fit?

The text has been changed to specify that we additionally calculate the composition without using the fit radii, but from the contrast function analysis.

In figure capture 2b we now state that the contrast error is shown in figure 2b as vertical lines. We also add that additional information about the contrast error can be found in the SI.

8) Line 138-139: the authors mention that the peak position shifts with changing the energy. If

this is coming from a structure factor, shouldn't this be energy independent? Indeed they write $S(q)$ and not $S(q,E)$. So, why the peak moves?

The Structure Factor is the same for all energies, but the Form Factor changes to such an extent that there is a shift in the location of the maximum scattering intensity. (This is the same reason that superimposing a linear function with a gaussian will shift the apparent maximum location of the gaussian)

Figure S12 has been added to show the Structure Factor and Form Factors separately. An addition has been made to line 158 in the text to refer the reader to Figure S12

9) Have the author tested that the poly soap (and the F127 as well) do not stick to the wall and cumulate during flow?

Yes, this was evaluated for both F127 and polysoap. We added a section to the SI called "Sample Integrity Considerations", which discusses this important question.

10) The part dealing with the pyrene uptake is not clearly presented. Is the poly soap as investigated by RSOXS loaded with pyrene or not?

If not, why reporting all the pyrene loading data?

If yes, please state this clearly and also what loading, etc.

We clarify that none of the RSOXS measurements presented had pyrene loading. The authors have begun investigating RSOXS of pyrene loaded micelles, but the results are not yet conclusive and are beyond the scope of this manuscript.

We modified the "Pyrene Uptake of aPS50" section of the SI to inform the reader that the pyrene uptake measurements validate aPS50 as being effective at its intended application of hydrocarbon sequestration.

11) in order to judge the quality and validity of the fit, I require the authors to present in the SI for some selected energies all the fitted components (structure and form factor independently).

For the F127, but especially for the poly soap, where the unchanging core dimension is claimed Figure S12 and S13 have been added to show the Structure Factor and Form Factors separates for figures 2a and 4c.

In summary, I believe that the manuscript reports a relevant study that is potential for publication in Nature Communication after the points raised by the reviewer are addressed.

We thank the reviewer for this positive statement regarding publication in Nature Communications once we address his points. We hope the editor agrees with us that all of the reviewer's points have been addressed and look forward to his decision.

Sincerely,

Brian A. Collins, Assistant Professor
Department of Physics and Astronomy
Washington State University

REVIEWERS' COMMENTS

Reviewer #3 (Remarks to the Author):

The reviewer is very satisfied with the author's answer.
I suggest publication in Nature Communication.

Prof. Giuseppe Portale